# FORWARD–BACKWARD FEATURE TRANSFER FOR INDUSTRIAL ANOMALY DETECTION AND SEGMENTATION

## ABSTRACT

Motivated by efficiency requirements, most industrial anomaly detection and segmentation (IADS) methods process low-resolution images, e.g., $224 \times 224$ pixels, obtained by downsampling the original input images. In this setting, downsampling is typically applied also to the provided ground-truth defect masks. Yet, as numerous industrial applications demand the identification of both large and small defects, this downsampling procedure may fail to reflect the actual performance achievable by current methods. In this work, we propose a fast approach based on a novel Teacher-Student paradigm. This paradigm relies on two shallow Student MLPs that learn to transfer patch features across the layers of a frozen Teacher Vision Transformer. Our framework can spot anomalies from high-resolution images faster than other methods, even when they process low-resolution images, achieving state-of-the-art overall performance on MVTec AD and segmentation results on VisA. We also propose novel evaluation metrics that capture robustness regarding defect size, i.e., the ability of a method to preserve good localization from large anomalies to tiny ones, focusing on segmentation performance as a function of anomaly size. Evaluating our method with these metrics reveals its stable performance in detecting anomalies of any size.

## 1 INTRODUCTION

Industrial anomaly detection and segmentation (IADS) aims to identify anomalous samples and localize their defects. This task is particularly challenging in industrial applications where anomalies are varied and unpredictable, and nominal samples may be scarce. In these settings, IADS is usually tackled in a *cold-start* fashion: the training procedure is unsupervised, with the train set comprising only images of nominal samples. Modern approaches for IADS Roth et al. (2022); Gudovskiy et al. (2022); Chiu & Lai (2023); Rudolph et al. (2023); Cao et al. (2022); Deng & Li (2022); Tien et al. (2023); Batzner et al. (2024) create a model of the nominal samples during training. Then, at inference time, each test sample is compared to this nominal model, and any discrepancy is interpreted as an anomaly. To reduce both training and inference time, all these IADS solutions process low-resolution images obtained by downsampling the original input images. However, this approach is detrimental to the task since smaller anomalies could be lost due to strong downsampling, as may be observed in Fig. 1. Moreover, it is common practice to downsample also the ground-truth defect masks provided with the benchmarks. Accordingly, as shown in Fig. 1, the areas of defects get smaller, and tiny anomalies may even disappear from the ground truth. Yet, since many industrial applications require the detection of both large and small defects, the practice described above might not accurately reflect the ability of current methods to localize defects of all sizes. Recently, EfficientAD Batzner et al. (2024) proposed a benchmark in which all the considered methods' outputs are upsampled to the original ground-truth resolution, although all methods still process a low-resolution input.

In this work, we propose a novel unsupervised IADS approach that can process high-resolution images faster than other methods, even when they process low-resolution images. This enables our technique to detect even smaller anomalies while maintaining applicability in industrial contexts. Our approach relies on a frozen Transformer backbone and a novel Teacher–Student paradigm whereby lightweight MLPs (i.e., the Students) shared across all patch embeddings learn to mimic the contextualization and decontextualization transformation occurring between the layers of the Transformer backbone (i.e., the Teacher) by observing only nominal samples. The core concept of our approach is that, after optimization, the Student networks can hallucinate contextual information from

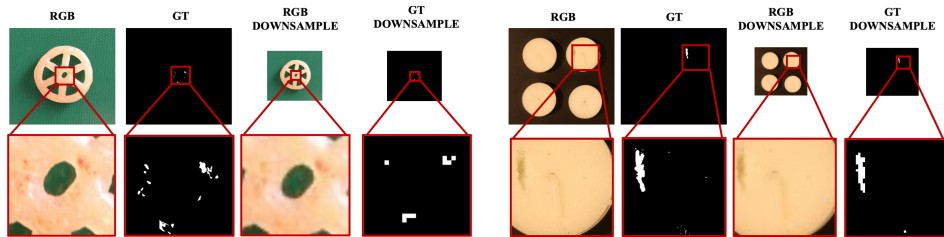

Figure 1: **Effects of downsampling on VisA**. Tiny defects are no longer visible in both RGB and GT.

more local content and vice versa, on nominal samples, and falter to do so in anomalous samples. At inference time, for each image patch, the actual features computed by the Teacher are compared to those predicted by the Students, with the discrepancies between the former and the latter highlighting the presence of anomalies.

Notably, our method formulation is general and can be applied to any Transformer feature extractor, as also supported by our experiments. However, by learning the contextualization and decontextualization pretext task on the feature extracted by DINO-v2 Oquab et al. (2023), which has been trained on images of varying resolutions, our approach achieves superior performance to other methods, as depicted in Fig. 2, even when trained and evaluated at high resolution, while being remarkably faster − ∼ 2 ms on a NVIDIA GeForce RTX 4090 to detect anomalies on $1036 \times 1036$ images.

The key to its speed is using shallow MLP student networks shared across patch features. In this way, each feature vector can be processed independently, allowing extremely fast batched processing. Moreover, each patch feature becomes a different training sample

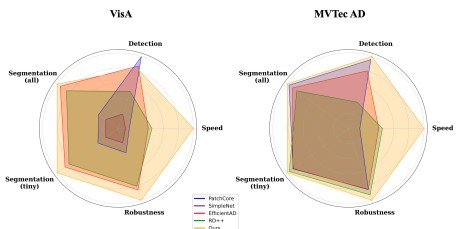

Figure 2: **Comparison between IADS methods.** The metrics reported in the charts are described in Section 4. Values are normalized for better readability.

for our Student networks, significantly enlarging the training set size compared to the number of training images. As a result, our method also achieves excellent few-shot performance.

Finally, to evaluate the advantages of processing high-resolution images, we propose novel evaluation metrics to assess the segmentation performance as a function of the size of the anomalies. This protocol captures the robustness concerning the defect size, i.e., the ability to preserve localization performance from large anomalies to smaller ones. Evaluating our method with this novel protocol revealed its ability to detect even tinier defects better than competitors.

Our contribution can be summarized as follows: *(i)* we propose a novel IADS method that exhibits state-of-the-art performance on MVTec AD and state-of-the-art segmentation performance on VisA, while running at a remarkably higher speed than all competitors; *(ii)* we introduce novel evaluation metrics to assess how effectively IADS methods handle anomalies of different sizes; *(iii)* we propose a challenging few-shot AD benchmark built upon the VisA dataset on which our proposal achieves state-of-the-art segmentation performance.

## 2 RELATED WORK

**Anomaly detection.** Several approaches have been proposed in the literature to perform IADS. These solutions can be categorized based on the approach followed to model nominal samples. Normalizing Flows Papamakarios et al. (2021); Yu et al. (2021); Rudolph et al. (2021); Gudovskiy et al. (2022); Chiu & Lai (2023) based methods construct complex distributions by transforming a probability density via a series of invertible mappings. In particular, these methods extract features of normal images from a pre-trained model and transform the feature distribution into a Gaussian distribution during the training phase. At test time, after passing the extracted features through the Normalizing Flow, the features of abnormal images will deviate from the Gaussian distribution of the training phase, suggesting an anomaly. Lately, several solutions Roth et al. (2022); Cohen & Hoshen

(2020); Bergman et al. (2020) that employ Memory Banks have been introduced. This category of solutions exploits well-known feature extractors trained on a large plethora of data Caron et al. (2021); Oquab et al. (2023); He et al. (2022) to model nominal samples. More in detail, during training, the feature extractor is kept frozen and used to compute features for nominal samples which are then stored in a memory bank. At test time, the features extracted from an input image are compared to those in the bank in order to identify anomalies. Despite their remarkable performance, these approaches suffer from slow inference speed, since each feature vector extracted from the input image needs to be compared against all the nominal feature vectors stored in the memory bank. Methods close to our solution which follows a Teacher–Student strategy Bergmann et al. (2020); Wang et al. (2021); Cao et al. (2022); Salehi et al. (2021); Deng & Li (2022); Batzner et al. (2024); Tien et al. (2023) have also been proposed. In this family of solutions, the training phase involves a Teacher model that extracts features from nominal samples and distils this knowledge to the Student model, which learns to mimic the Teacher's feature extraction process. During the testing phase, differences between the features generated by the Teacher model and those produced by the Student model reveal the presence of anomalies. Recently, a multimodal approach Costanzino et al. (2024) investigated the idea of mapping features from one modality to the other on nominal samples and then detecting anomalies by pinpointing inconsistencies between observed and mapped features. This solution leverages MLPs to learn a mapping between features coming from two different modalities, RGB images and point clouds. Conversely, our novel solution does not require two modalities.

**Anomaly detection datasets.**    During the last few years, several IADS datasets have been released. The introduction of MVTec AD Bergmann et al. (2019) kicked off the development of IADS approaches for industrial applications. This dataset contains several industrial inspection scenarios, each comprising train and test sets. Each train set contains only nominal images, while the test sets also contain anomalous samples. Such a scenario represents realistic real-world applications where types and possible locations of defects are unknown during the development of IADS algorithms. Later, the work was extended with the MVTec 3D-AD Bergmann et al. (2022b) dataset, which follows the same structure of MVTec AD, but also provides the pixel-aligned point clouds of the samples to address the IADS in a multimodal fashion. Shortly afterward, the Eyecandies Bonfiglioli et al. (2022) dataset was released, miming the structure of MVTec 3D-AD by introducing a multimodal synthetic dataset containing images, point clouds, and normals for each sample. To provide a more challenging scenario the VisA dataset Zou et al. (2022) has been introduced, in which high-resolution images of complex scenes that can also contain multiple instances of the same object have been released. In the end, more task-specific datasets such as MAD Zhou et al. (2023) and MVTec LOCO Bergmann et al. (2022a) have been released. In particular, MAD Zhou et al. (2023) introduced a multi-pose dataset with images from different viewpoints covering a wide range of poses to tackle a pose-agnostic IADS. MVTec LOCO Bergmann et al. (2022a) contains not only structural anomalies, such as dents or holes but also logical anomalies, which violations of logical constraints can be for instance a wrong ordering or a wrong combination of normal objects.

## 3  METHOD

As outlined in  Fig. 3, our method follows a Teacher-Student paradigm in which the Teacher, $\mathcal{T}$, is a frozen Transformer encoder (e.g., DiNO-v2 Oquab et al. (2023)), while the two Students, referred to as Forward and Backward Transfer Networks ($\mathcal{S}_F$ and $\mathcal{S}_B$) are realized as shallow MLPs.

**Overview.**    The Students are trained on nominal samples and learn to mimic the transformations between the patch embeddings occurring within the layers of the Transformer. In particular, the Forward Transfer Network learns to predict the patch embeddings computed by a layer of the Transformer ($k$ in  Fig. 3), given the corresponding embeddings computed by a previous layer ($j$ in Fig. 3). Conversely, the Backward Transfer Network learns to predict the features calculated by the Transformer at layer $j$ given the corresponding ones at layer $k$. The Student networks $\mathcal{S}_F$, $\mathcal{S}_B$ are shared across patch embeddings, i.e., both take as input the features associated with the patch $(i)$ at a layer $f_j^{(i)}$, $f_k^{(i)}$ and predict the corresponding features at the other layer $\hat{f}_k^{(i)}$, $\hat{f}_j^{(i)}$. At inference time, for all patch embeddings of the given test sample, the features predicted by the Students are compared to the ones extracted by the Teacher, with the discrepancies between the former and the latter providing the signal to highlight anomalies. As shown in  Fig. 3, the difference between the outputs from $\mathcal{S}_F$, $\mathcal{S}_B$ and the patch embeddings from layers $k$,$j$ of $\mathcal{T}$ yield two anomaly maps,

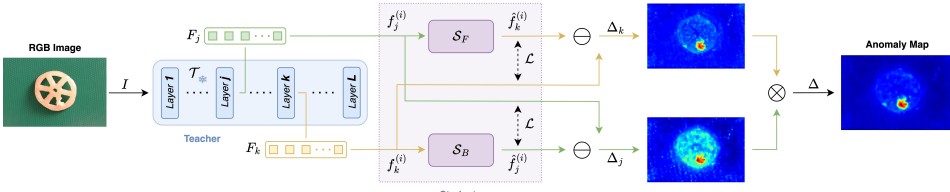

Figure 3: **FBFT overview.** Given an RGB Image $I$, a frozen pre-trained transformer backbone $\mathcal{T}$ is leveraged to extract two sets of patch-aligned features $F_j, F_k$, from different layers, one from a lower contextualization layer $j$ and one from a higher contextualization layer $k$. Then, a pair of feature transfer networks, $\mathcal{S}_F, \mathcal{S}_B$, predict the extracted features from one layer to the other, processing the features at each patch independently. Lastly, extracted and transferred features are compared through a Euclidean distance, to create contextualization-specific anomaly maps, $\Delta_j, \Delta_k$, that are then combined to obtain the final anomaly map $\Delta$.

$\Delta_j$ and $\Delta_k$, that are fused to obtain the final one. Due to the novel pretext tasks employed by our approach, realized through $\mathcal{S}_F, \mathcal{S}_B$, we dub it Forward Backward Feature Transfer (FBFT).

**Rationale.** The intuition behind our approach relies on the observation that as patch embeddings travel from shallower to deeper layers of a Transformer encoder, they become increasingly contextualized, i.e., deeper representations capture more global information that helps singling out a patch based on the specific context provided by the input image.

Our Forward and Backward Transfer networks are trained to contextualize and decontextualize patch embeddings according to the function, which we assume to be invertible, executed by the Transformer between a pair of chosen layers. In particular, contextualization gathers and integrates local details to form a coherent global understanding of an image; conversely, decontextualization is the opposite of this process, i.e., finding local features from a global understanding of the image.

By learning contextualization and decontextualization on nominal samples, our Students will understand how local features, such as edges and textures, transform into larger structures, such as shapes and objects, for normal entities. However, when presented with anomalous samples, this mapping breaks down because the predicted local and global features do not align with those extracted by the Teacher model, revealing inconsistencies that indicate anomalies.

Moreover, we conjecture that feature contextualization and decontextualization are complex functions that do not admit a trivial solution, such as, e.g., the identity function. Therefore, small-capacity networks trained only on nominal samples are unlikely to learn general functions that can yield correct predictions on out-of-distribution data, i.e., features extracted from anomalous patches.

**Teacher.** As a first step, we provide as input to the Teacher $\mathcal{T}$ an image $I$ with dimensions $H \times W \times C$, where $H$, $W$, and $C$ correspond to the height, width, and number of channels. In our framework, we employ a Transformer-based backbone that provides a set of features, one for each input patch processed by the backbone after each layer. Each feature, $f^{(i)} \in \mathbb{R}^D$, has dimension $D$ according to the inner representation employed by the backbone, while the number of features is $N = HW/P^2$, where the patch size is $P \times P$ pixels. During the forward pass, we extract two sets of features, $F_j = \left\{ f_j^{(i)}, i = 1 \cdots N \right\}$ and $F_k = \left\{ f_k^{(i)}, i = 1 \cdots N \right\}$, from two different layers of the backbone, i.e. layers $j$ and $k$, with $j < k$.

We highlight that, as far as the representation of small defects is concerned, a Transformer backbone can effectively handle high-resolution inputs because, although it processes images by dividing them into patches, which results in smaller spatial size, the input information is not compressed, on the contrary, each patch is expanded to a higher dimensionality related to the internal representation of the Transformer (e.g., RGB patches of $14 \times 14 \times 3$ pixels are mapped into 768-dimensional embeddings). Therefore, as high resolution information is retained, we can also detect smaller defects.

**Students.** The two sets of features extracted by the Teacher are processed by a pair of Forward and Backward Transfer networks, $\mathcal{S}_F$ and $\mathcal{S}_B$, representing the Students in our architecture. $\mathcal{S}_F$ maps

a feature vector from a less contextualized layer $j$ to a more contextualized layer $k$, while $\mathcal{S}_B$ does the opposite. Each network predicts the features of one layer from the corresponding ones extracted from the other, processing each patch location independently. Thus, given a patch location $(i)$ and the corresponding features $f_j^{(i)}$ and $f_k^{(i)}$, the features predicted by the Students can be expressed as:

$$\hat{f}_k^{(i)} = \mathcal{S}_F(f_j^{(i)}) \quad \hat{f}_j^{(i)} = \mathcal{S}_B(f_k^{(i)}) \tag{1}$$

where $\mathcal{S}_F$ and $\mathcal{S}_B$ are parametrized as MLPs, shared across all patches. By processing all patches, we obtain the two sets of transferred features: $\hat{F}_j = \left\{ \hat{f}_j^{(i)}, i = 1 \cdots N \right\}$ and $\hat{F}_k = \left\{ \hat{f}_k^{(i)}, i = 1 \cdots N \right\}$.

As stated in Section 1, employing Student networks that process each patch independently with shallow MLPs enables fast batched inference. Moreover, as each patch is in an independent training sample, this approach effectively increases the training set size relative to the number of training images. Consequently, our method can be trained on a few images while achieving excellent performance (see Section 5).

**Training.** During training, the weights of $\mathcal{S}_F$ and $\mathcal{S}_B$ are optimized only on nominal samples of a specific class from a dataset. For both networks, we employ the cosine distance between the features extracted from the backbone at the considered layers and the transferred ones as a loss function. More details on the employed loss can be found in Appendix A.2. Thus, the per-patch losses are:

$$\mathcal{L}_j^{(i)}\left(f_j^{(i)}, \hat{f}_j^{(i)}\right) = 1 - \frac{f_j^{(i)} \cdot \hat{f}_j^{(i)}}{\|f_j^{(i)}\|\|\hat{f}_j^{(i)}\|} \quad \mathcal{L}_k^{(i)}\left(f_k^{(i)}, \hat{f}_k^{(i)}\right) = 1 - \frac{f_k^{(i)} \cdot \hat{f}_k^{(i)}}{\|f_k^{(i)}\|\|\hat{f}_k^{(i)}\|} \tag{2}$$

**Inference.** At inference time, the image under analysis is processed by the Transformer backbone and the features extracted from the two layers, $F_j$ and $F_k$ are provided as input to the Forward and Backward Transfer networks to obtain the corresponding transferred features, $\hat{F}_j$ and $\hat{F}_k$. The Euclidean distance is then employed to compute the patch-wise differences between extracted and transferred features $\Delta_j^{(i)}, \Delta_k^{(i)}$:

$$\Delta_j^{(i)} = \|f_j^{(i)} - \hat{f}_j^{(i)}\|_2 \quad \Delta_k^{(i)} = \|f_k^{(i)} - \hat{f}_k^{(i)}\|_2, \quad i = 1 \ldots N \tag{3}$$

Typically, we can identify anomalies from both $\Delta_j^{(i)} \Delta_j^{(j)}$, i.e., from both transfer directions. However, in case of failure of the Student networks, the bidirectional mapping creates a fail-safe mechanism since it is unlikely for an anomaly to pass through contextualization and decontextualization without detection. Thus, we fuse the predicted anomaly maps $\Delta_j^{(i)}$ and $\Delta_j^{(j)}$ by multiplying those corresponding to the same patch:

$$\Delta^{(i)} = \Delta_j^{(i)} \cdot \Delta_k^{(i)}, \quad i = 1 \ldots N \tag{4}$$

This fusion strategy let us achieve more accurate results, as shown in Table 6 of Appendix. More details on the employed fusion function in Appendix A.3.

Finally, the set of fused differences, $\Delta^{(i)}$, is reshaped as a $\sqrt{N} \times \sqrt{N}$ anomaly map according to the positions of the patches within the input image. This map is then up-sampled to $H \times W$, i.e. the spatial size of the input image, by bilinear interpolation and successively smoothed according to common practice Roth et al. (2022); Costanzino et al. (2024); Tien et al. (2023); Liu et al. (2023). The global anomaly score required to perform sample-level anomaly detection is computed as the mean value of the top $M$ values of the final anomaly map $\Delta$.

## 4 EXPERIMENTAL SETTINGS

### 4.1 DATASETS

To assess our proposal we rely on two IADS datasets: VisA Zou et al. (2022) and MVTec AD Bergmann et al. (2019). The VisA Zou et al. (2022) dataset provides images of varying resolution, with the height spanning from 1284 to 1562 pixels and anomalies as tiny as 1 pixel and up to 478781 pixels. The dataset contains 10821 images of 12 objects across 3 domains, with challenging scenarios including complex structures in objects, multiple instances, and pose variations. Between the provided images, 9621 are nominals while 1200 contains defects.

The MVTec AD dataset mimics real-world industrial inspection scenarios and includes 5354 images, with heights spanning from 700 to 1024 pixels and anomalies ranging from 24 pixels to 517163 pixels. The images pertain to 15 objects exhibiting 73 different types of anomalies for 1888 anomalous samples. Both Visa and MVTec AD provide pixel-accurate ground truths for each anomalous sample.

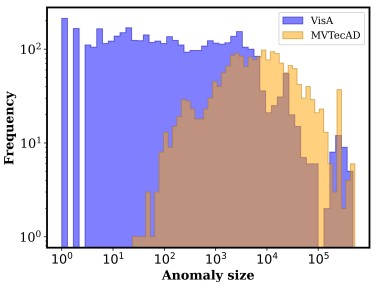

Figure 4: **Anomaly size distribution.**

As highlighted in Fig. 4, VisA features a significantly wider range of anomaly sizes and includes tiny defects. As a result, downsampling the ground-truths to $224 \times 224$ pixels, i.e., the most commonly employed inference and evaluation size in present literature, yields a reduction in the number of defects of 21.42% and 0.37% for VisA and MVTec AD, respectively. These observations render VisA a particularly challenging scenario for assessing the robustness of IADS methods with respect to defect size.

### 4.2 METRICS

**Standard Metrics.** We utilize the metrics employed in MVTec AD Bergmann et al. (2019) and VisA Zou et al. (2022). These two datasets assess the image anomaly detection performance employing the Area Under the Receiver Operator Curve (I-AUROC) computed on the global anomaly score. As for segmentation performance, the Area Under the Per-Region Overlap (AUPRO) on the anomaly map is computed, with the integration threshold set to 0.3. Recently, Batzner et al. (2024); Costanzino et al. (2024) have proposed to compute the AUPRO considering a tighter threshold, i.e., 0.05. We will consider both metrics and denote AUPROs with integration thresholds 0.3 and 0.05 as AUPRO@30%, and AUPRO@5%, respectively.

**Performance across defects sizes.** To highlight the capability of each method to segment defects with varying sizes, we introduce a variation of the AUPRO metric. In particular, for each object in a dataset, we compute the anomaly size distribution and partition it in cumulative quartiles, denoted as $Q_1, Q_2, Q_3, Q_4$. These cumulative quartiles are associated with sets that contain only anomalies with a size smaller than or equal to the considered quartile. Hence, the set associated with Q4 consists of all anomalies, while Q1 includes only the smallest ones. Then, we calculate the AUPRO@30% and AUPRO@5% on each set, with the segmentation metrics associated with Q4 being the already described segmentation metrics adopted in the standard benchmarks.

**Robustness.** We also introduce a novel metric, $\rho$, to assess the robustness of a method w.r.t. the size of the defects in a dataset. In particular, $\rho$ captures a method's ability to segment tiny and larger defects accurately. Accordingly, we define the robustness as:

$$\rho = w \cdot (1 - s), \quad s = \frac{|\text{AUPRO}(Q_4) - \text{AUPRO}(Q_1)|}{\max(\text{AUPRO}(Q_1), \text{AUPRO}(Q_4))}, \quad w = \frac{1}{4} \cdot \sum_{i=1}^{4} \text{AUPRO}(Q_i) \quad (5)$$

Here, for the sake of compactness, we denote as AUPRO either AUPRO@5% or AUPRO@30%, such that considering the former or the latter will yield $\rho$@5% or $\rho$@30%, respectively. In the definition of $\rho$, the AUPRO is evaluated for the smallest defects only, i.e., $\text{AUPRO}(Q_1)$, and for all defects, i.e., $\text{AUPRO}(Q_4)$. With this measure, if a method can correctly segment larger defects but struggles with small ones, its sensitivity to defect size, $s$, is high and its robustness, $\rho$, is low. Conversely, a robust method should be able to accurately segment defects regardless of their sizes, which in our metric would be captured by the difference between $\text{AUPRO}(Q_4)$ and $\text{AUPRO}(Q_1)$ turning out low, yielding low sensitivity and high robustness. Yet, to avoid deeming as robust a method that performs poorly on both small and large defects, such that $\text{AUPRO}(Q_4)$ and $\text{AUPRO}(Q_1)$ are both similarly low, we propose to introduce the average AUPRO across all quartiles, denoted as $w$, as weighing factor of the term $(1 - s)$ in the definition of $\rho$. It is worth pointing out that the proposed robustness metric, $\rho$, is bounded by 1 since both $s$ and $w$ are smaller than 1.

### 4.3 Evaluation Protocol and Implementation Details

**Evaluation Protocol.** We evaluate our proposal, FBFT, alongside with several state-of-the-art IADS methods, such as PatchCore Roth et al. (2022), SimpleNet Liu et al. (2023), EfficientAD Batzner et al. (2024) and RD++ Tien et al. (2023). EfficientAD Batzner et al. (2024) proposes two variants: EfficientAD-S and EfficientAD-M. We consider the latter since it provides better IADS performance.

As described in Batzner et al. (2024), the results reported in SimpleNet Liu et al. (2023) are obtained by repeatedly evaluating the metrics on all test images during training to select the best check-point. Analyzing the official implementation, we noticed how this protocol has been followed also by RD++ Tien et al. (2023). However, in real-world settings, the test data is not available at training time. Thus, to avoid overestimating the actual performance of the models, we disable the above check-point selection mechanism, train all methods for a fixed number of epochs and evaluate the model obtained at the last checkpoint. For Batzner et al. (2024); Liu et al. (2023); Tien et al. (2023), we train for the number of epochs specified in the official implementations.

PatchCore Roth et al. (2022) employs a centre-crop of the input images since in MVTec AD, most of the defects lie within this cropped area. However, in a real-world scenario, anomalies can occur outside of this area, thus, we disable this strategy as it implies knowledge about the location of anomalies in the test set.

As anticipated in Section 1, we compute all metrics based on the original ground-truths provided with the datasets, which have the same resolution as the original input images. Hence, we do not downsample the ground-truths to the input image size processed by a method, but we bilinearly upsample the anomaly map to the same resolution as the ground-truth in order to calculate all metrics.

Some methods, including ours, must add padding to the input image to adapt it to the input size of the employed backbone. However, we remove these extra pixels from the final anomaly maps as, otherwise, they usually decrease the False Positive Rate (and thus artificially ameliorate the segmentation metrics) because they tend to yield very low anomaly scores. Finally, we calculate the AUPRO considering all the samples in the test set, both nominal and anomalous [1].

**Implementation details.** As our default Teacher network, we employ DINO-v2 ViT-B/14 Oquab et al. (2023) pre-trained on a large, curated, and diverse dataset of 142 million images, comprising ImageNet-22k Deng et al. (2009); Ridnik et al. (2021). Thus, our $\mathcal{T}$ network processes $1036 \times 1036 \times 3$ RGB images and outputs $74 \times 74 \times 768$ feature maps. Both $\mathcal{S}_F$ and $\mathcal{S}_B$ consist of three linear layers, each but the last one followed by GeLU activations. The number of units per layer is 768 for both $\mathcal{S}_F$ and $\mathcal{S}_B$. The two networks are trained jointly for 50 epochs using Adam Kingma & Ba (2015) with a learning rate of 0.001. As default, we select the layers $j = 8$ and $k = 12$ to realize the Feature Transfer Networks. A detailed ablation study on the choice of the best pair of layers is reported in Appendix A.1. We employed $M = 0.001 \cdot H \cdot W$ to attain the number of pixels used to calculate the global anomaly score. We conducted all the experiments on a single NVIDIA GeForce RTX 4090.

## 5 Experiments

**Anomaly detection and segmentation.** For a fair evaluation, for both training and inference, we provide input to all methods images at the highest resolution that would enable execution on a single GPU to avoid or minimize downsampling. In particular, we could handle input images up to $1036 \times 1036$ pixels with EfficientAD, RD++, and FBFT, while the highest input resolution for PatchCore and SimpleNet was found to be $512 \times 512$ pixels. The anomaly detection and segmentation results on VisA and MVTec AD are reported in Table 1. Our approach achieves the best segmentation results on the VisA dataset, with 0.952 AUPRO@30% and 0.787 AUPRO@5% and the state-of-the-art in both detection and segmentation on the MVTec AD dataset, with 0.988 I-AUROC, 0.945 AUPRO@30%, and 0.782 AUPRO@5% Regarding detection performance on VisA, our method attains results comparable to the runner-up (0.968 of EfficientAD vs. 0.964 of Ours). The supplemental material provides the detailed per-class metrics for each method. In Fig. 5, we depict some qualitative results on the VisA dataset. Our method provides more localized anomaly scores

---

[1]We noticed that official code from Roth et al. (2022), calculates the AUPRO only on anomalous test samples, obtaining higher scores since the false positive rate is inherently lower with this protocol.

Table 1: **I-AUROC, AUPRO30@% and AUPRO5@% on VisA and MVTec AD for several IADS methods.** Average metrics of all classes on the respective test set. Best results in **bold**, runner-ups underlined. All methods are trained and tested at high resolution.

| ALGORITHM | VisA | | | MVTec AD | | |
|---|---|---|---|---|---|---|
| | I-AUROC | AUPRO@30% | AUPRO@5% | I-AUROC | AUPRO@30% | AUPRO@5% |
| PatchCore | **0.982** | 0.752 | 0.542 | 0.983 | 0.937 | 0.701 |
| SimpleNet | 0.904 | 0.718 | 0.469 | – | – | – |
| EfficientAD | 0.968 | 0.937 | 0.777 | 0.965 | 0.920 | 0.757 |
| RD++ | 0.930 | 0.907 | 0.758 | 0.915 | 0.901 | 0.716 |
| FBFT (Ours) | 0.964 | **0.952** | **0.787** | **0.988** | **0.945** | **0.782** |

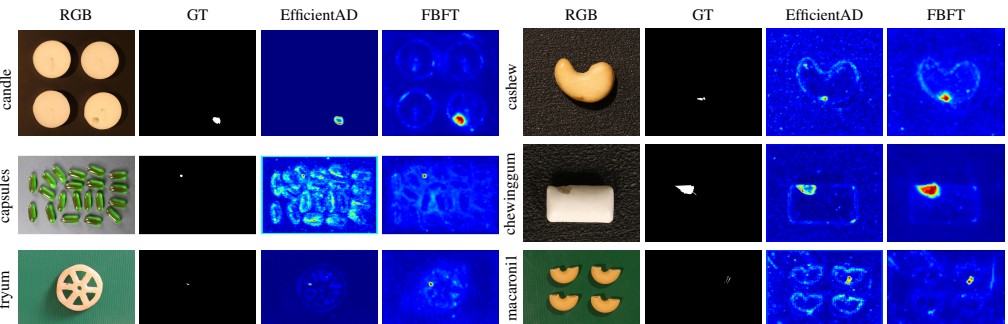

Figure 5: **VisA dataset qualitative results.** All methods are trained and tested at high resolution.

compared to EfficientAD Batzner et al. (2024), i.e., the second-best method on VisA. For instance, by looking at the `capsules` example, our anomaly score peak is centered on the anomaly differently from Batzner et al. (2024). Further qualitative results are reported in Appendix A.12.

**Cumulative quartiles based anomaly segmentation.** We report in Table 2 the analysis on VisA and MVTec AD of the performance w.r.t. anomaly size using the cumulative quartile metrics defined in Section 4. The results highlight that the defect size impacts the segmentation metrics, especially for the tiniest ones, i.e., the anomalies in $Q_1$. Nevertheless, our method is the best across all quartiles, with a notable gap compared to the second-best method on $Q_1$ on VisA, which is the dataset with the highest frequency of tiny defects (e.g., AUPRO30@% 0.935 Ours vs. 0.890 EfficientAD). Moreover, our method is remarkably stable and robust across quartiles. For instance, on VisA, we go from 0.758 to 0.730 AUPRO5@%, losing only 2.6% segmentation quality, much less than the runner-up method, EfficientAD, which decreases its performance of 6.7%, from 0.743 to 0.693 AUPRO5@%.

**Inference time and input resolution ablation.** We report in Table 3 the inference time and main IADS metrics on VisA for our method and state-of-the-art approaches Roth et al. (2022); Liu et al. (2023); Batzner et al. (2024); Tien et al. (2023). Using the same machine, we compute the speed in `ms` per sample as the average across all the test samples of the VisA dataset. For each method, we compute

Table 2: **Quartile-based segmentation metrics.** Best results in **bold**, runner-ups underlined. Results on VisA (top) and MVTec AD (bottom). All methods are trained and tested at high resolution.

| ALGORITHM | DATASET | AUPRO@30% | | | | | | AUPRO@5% | | | | | |
|---|---|---|---|---|---|---|---|---|---|---|---|---|---|
| | | $Q_1$ | $Q_2$ | $Q_3$ | $Q_4$ | $\overline{Q}$ | $\rho$@30% | $Q_1$ | $Q_2$ | $Q_3$ | $Q_4$ | $\overline{Q}$ | $\rho$@5% |
| PatchCore | | 0.703 | 0.720 | 0.740 | 0.752 | 0.728 | 0.679 | 0.484 | 0.492 | 0.518 | 0.542 | 0.509 | 0.454 |
| SimpleNet | | 0.658 | 0.668 | 0.696 | 0.718 | 0.685 | 0.627 | 0.390 | 0.399 | 0.435 | 0.469 | 0.423 | 0.351 |
| EfficientAD | VisA | 0.890 | 0.923 | 0.933 | 0.937 | 0.920 | 0.873 | 0.693 | 0.741 | 0.763 | 0.777 | 0.743 | 0.662 |
| RD++ | | 0.867 | 0.898 | 0.906 | 0.907 | 0.894 | 0.853 | 0.710 | 0.740 | 0.755 | 0.758 | 0.740 | 0.692 |
| FBFT (Ours) | | **0.935** | **0.941** | **0.946** | **0.952** | **0.943** | **0.926** | **0.730** | **0.749** | **0.768** | **0.787** | **0.758** | **0.702** |
| PatchCore | | 0.924 | 0.932 | 0.935 | 0.937 | 0.932 | 0.918 | 0.653 | 0.677 | 0.691 | 0.701 | 0.680 | 0.633 |
| EfficientAD | MVTec AD | 0.922 | 0.925 | 0.925 | 0.920 | 0.923 | 0.920 | 0.758 | 0.769 | 0.767 | 0.757 | 0.762 | 0.760 |
| RD++ | | 0.946 | 0.922 | 0.918 | 0.901 | 0.921 | 0.952 | **0.782** | 0.752 | 0.744 | 0.716 | 0.748 | 0.684 |
| FBFT (Ours) | | **0.958** | **0.948** | **0.947** | **0.945** | **0.949** | **0.986** | **0.806** | **0.798** | **0.795** | **0.782** | **0.795** | **0.783** |

Table 3: **Performance and inference time on VisA at different input resolution.** Inference time in `ms` per sample. Best results in **bold**, runner-ups underlined.

| ALGORITHM | INPUT RESOLUTION | INFERENCE TIME | I-AUROC | AUPRO@30% | | | | | | AUPRO@5% | | | | | |
|---|---|---|---|---|---|---|---|---|---|---|---|---|---|---|---|
| | | | | $Q_1$ | $Q_2$ | $Q_3$ | $Q_4$ | $\overline{Q}$ | $\rho$@30% | $Q_1$ | $Q_2$ | $Q_3$ | $Q_4$ | $\overline{Q}$ | $\rho$@5% |
| PatchCore | | 87.151 | 0.948 | 0.739 | 0.741 | 0.760 | 0.779 | 0.754 | 0.715 | 0.443 | 0.441 | 0.471 | 0.508 | 0.465 | 0.405 |
| SimpleNet | | 210.833 | 0.896 | 0.650 | 0.654 | 0.671 | 0.690 | 0.666 | 0.627 | 0.309 | 0.311 | 0.338 | 0.372 | 0.332 | 0.275 |
| EfficientAD | Original | 7.837 | **0.984** | 0.876 | 0.904 | 0.919 | 0.931 | 0.907 | 0.853 | 0.646 | 0.663 | 0.697 | 0.732 | 0.684 | 0.603 |
| RD++ | | 17.748 | 0.856 | 0.770 | 0.787 | 0.814 | 0.843 | 0.803 | 0.733 | 0.411 | 0.429 | 0.478 | 0.541 | 0.464 | 0.352 |
| FBFT (Ours) | | 1.321 | 0.938 | 0.809 | 0.815 | 0.822 | 0.831 | 0.819 | 0.787 | 0.652 | 0.658 | 0.667 | 0.688 | 0.666 | 0.700 |
| PatchCore | | 227.230 | 0.982 | 0.703 | 0.720 | 0.740 | 0.752 | 0.728 | 0.679 | 0.484 | 0.492 | 0.518 | 0.542 | 0.509 | 0.454 |
| SimpleNet | | 560.17 | 0.896 | 0.658 | 0.668 | 0.696 | 0.718 | 0.685 | 0.627 | 0.390 | 0.399 | 0.435 | 0.469 | 0.423 | 0.351 |
| EfficientAD | High | 82.367 | 0.968 | 0.890 | 0.923 | 0.933 | 0.937 | 0.920 | 0.873 | 0.693 | 0.741 | 0.763 | 0.777 | 0.743 | 0.662 |
| RD++ | | 63.176 | 0.930 | 0.867 | 0.898 | 0.906 | 0.907 | 0.894 | 0.853 | 0.710 | 0.740 | 0.755 | 0.758 | 0.740 | 0.692 |
| FBFT (Ours) | | **1.786** | 0.964 | **0.935** | **0.941** | **0.946** | **0.952** | **0.943** | **0.926** | **0.730** | **0.749** | **0.768** | **0.787** | **0.758** | **0.702** |

the inference time, from when the sample is available on the GPU to the computation of the anomaly scores, after a GPU warm-up, synchronizing all threads before estimating the total inference time. Our approach attains state-of-the-art anomaly segmentation performance, namely AUPRO@30%, $\rho$@30%, AUPRO@5%, and $\rho$@5%, while being extremely fast. We highlight that, even though PatchCore attains the best detection performance on the VisA dataset, it largely falls behind in terms of segmentation performance (AUPRO@30%=0.752 of PatchCore vs. AUPRO@30%=0.952 of Ours, AUPRO@5%=0.542 of PatchCore vs. AUPRO@5%=0.787 of Ours), and inference speed (227.230 `ms` of PatchCore vs. 1.786 `ms` of Ours).

We also include the results obtained by evaluating each competitor using inputs at their official low resolution (e.g., $224 \times 224$), reporting the performance for each anomaly size quartile, following the evaluation protocol described in Section 4.3. We also report the results of FBFT when trained and evaluated at $224 \times 224$. Comparing each method across different resolutions, we note how segmentation performance typically drops in all metrics when using the official input resolution, especially when detecting tiny anomalies, e.g., for RD++, from 0.710 to 0.411 in Q1 for AUPRO@5%. We highlight that FBFT processing high-resolution images is the best on all metrics. These results emphasize the key role of input resolution in maintaining consistent segmentation across defect sizes.

**Few-shot anomaly detection and segmentation.**   As mentioned in Section 1, collecting many nominal samples in most industrial scenarios can be extremely expensive or unfeasible. Also, frequent production changeover requires fast adaptation. For these reasons, a beneficial property of IADS methods is the ability to create a model of the nominal data even with few samples. We define a few-shot benchmark – based on the VisA dataset – randomly selecting 5, 10, and 50 images from each category as training data. We train the competitors Roth et al. (2022); Liu et al. (2023); Batzner et al. (2024); Tien et al. (2023) along with our proposed approach on these samples, and we test them on the entire test set, with the evaluation protocol proposed in Section 4, reporting the results in Table 4. We obtain the best segmentation performance for both metrics (AUPRO@30% and AUPRO@5%) in all the few-shot settings, significantly improving the most challenging segmentation metrics (+0.167 AUPRO@5% 5-shot) and retaining a stable segmentation performance (AUPRO@30% always above 0.9) across the various settings. These results confirm the ability of our method to optimize feature transfer networks even from a few nominal samples, thanks to the patch–independent processing enabled by the MLPs.

Table 4: **Few-shot IADS performance.** Best results in **bold**, runner-ups underlined.

| ALGORITHM | Full | | | 50-shot | | | 10-shot | | | 5-shot | | |
|---|---|---|---|---|---|---|---|---|---|---|---|---|
| | I-AUROC | AUPRO@30% | AUPRO@5% | I-AUROC | AUPRO@30% | AUPRO@5% | I-AUROC | AUPRO@30% | AUPRO@5% | I-AUROC | AUPRO@30% | AUPRO@5% |
| PatchCore | **0.982** | 0.752 | 0.542 | **0.959** | 0.724 | 0.485 | **0.948** | 0.704 | 0.459 | **0.916** | 0.698 | 0.455 |
| SimpleNet | 0.896 | 0.718 | 0.469 | 0.917 | 0.758 | 0.430 | 0.883 | 0.725 | 0.408 | 0.862 | 0.691 | 0.377 |
| EfficientAD | 0.968 | 0.937 | 0.777 | 0.831 | 0.854 | 0.569 | 0.816 | 0.806 | 0.469 | 0.810 | 0.834 | 0.511 |
| RD++ | 0.930 | 0.907 | 0.758 | 0.776 | 0.861 | 0.563 | 0.615 | 0.733 | 0.303 | 0.555 | 0.654 | 0.253 |
| FBFT (Ours) | 0.964 | **0.952** | **0.787** | 0.927 | **0.934** | **0.743** | 0.897 | **0.910** | **0.695** | 0.879 | **0.901** | **0.678** |

**Backbone ablation.**   All previous results were achieved using FBFT with DINO-v2 as the Teacher, $\mathcal{T}$, backbone. However, since our method formulation is general, we also explored different Transformer backbones, such as ViT/B-16 pre-trained on ImageNet, reporting segmentation results on ViSA in row 5 of Table 5. This network was pre-trained on $224 \times 224$ images with a classification objective. Thus, we resize images at $224 \times 224$ at inference time. We observe a performance drop compared to FBFT when processing high-resolution images (last row) with DINO-v2. Despite this,

Table 5: **Segmentation metrics on VisA.** Best results in **bold**, runner-ups underlined.

| ALGORITHM | BACKBONE | INPUT RESOLUTION | AUPRO@30% | AUPRO@5% |
|---|---|---|---|---|
| PatchCore | WideResNet101 | $224 \times 224$ | 0.779 | 0.508 |
| SimpleNet | WideResNet50 | $224 \times 224$ | 0.690 | 0.372 |
| EfficientAD | Custom | $224 \times 224$ | 0.931 | 0.732 |
| RD++ | WideResNet50 | $224 \times 224$ | 0.843 | 0.541 |
| FBFT | ViT/B-16 | $224 \times 224$ | 0.868 | 0.610 |
| FBFT | DINO-v2 | $224 \times 224$ | 0.831 | 0.688 |
| FBFT | DINO-v2 | $1036 \times 1036$ | **0.952** | **0.787** |

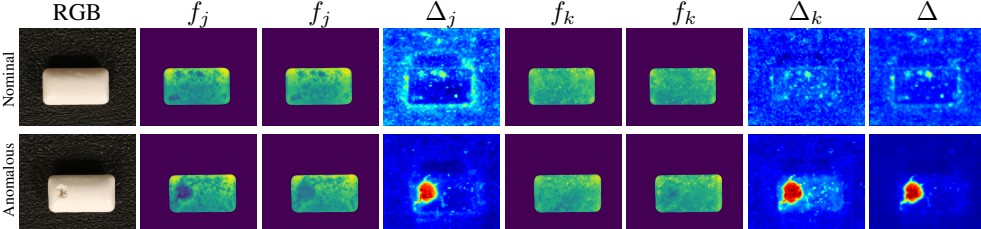

Figure 6: **Features visualization**. Channels average of feature maps before and after feature mapping.

our FBFT method outperforms all competitors except EfficientAD when operating on $224 \times 224$ resolution images. This demonstrates that our method can deliver competitive results with different backbones, although optimal performance is achieved by processing high-resolution images.

One could attribute FBFT's superior performance to the larger dataset used for pre-training DINO-v2 ($\sim 142$M images), compared to ImageNet ($\sim 14$M images), which was used for training WideResNet101 and ViT/B-16. However, if we compare FBFT performance with DINO-v2 at $224 \times 224$ resolution to DINO-v2 at $1036 \times 1036$ (last vs second-to-last rows of Table 5), we note a significant drop in performance, sometimes larger than the drop obtained when using ViT-B/16. This suggests that the excellent performance is primarily driven by high-resolution image processing rather than the choice of the pre-trained backbone.

**Features visualization.** In Fig. 6, we show the contextualized feature maps before $f_j$, $f_k$ and after $\hat{f}_j$, $\hat{f}_k$ the feature transfer, as well as their $\Delta_j$, $\Delta_k$ and final anomaly maps $\Delta$, for a nominal (top) and an anomalous (bottom) test sample of VisA. In the nominal case, we can notice how the features before and after the feature transfer look similar, resulting in low anomaly scores. In the anomalous case, as the extracted features $f_j$, $f_k$ fall out of the nominal distribution, the feature transfer network fails to contextualize or decontextualize them, resulting in erroneously reconstructed features $\hat{f}_j$ and $\hat{f}_k$. Thus, by analyzing the discrepancy between the original and reconstructed features, we produce accurate anomaly maps. Furthermore, after the combination, the overall anomaly map $\Delta$ exhibits less noise, thanks to the product-based aggregation employed in this work.

## 6 DISCUSSION

We introduced a fast approach based on Forward–Backward Feature Transfer that processes features extracted from layers with different contextualization levels of a Transformer backbone. We devised a novel metric to evaluate the stability of existing methods in segmenting anomalies of different sizes, spanning from very tiny to larger ones, along with a fair and sound training and evaluation protocol to assess the performance. The proposed solution achieves the best segmentation results on the VisA dataset, both on the classical benchmark and the proposed novel metrics, while running remarkably faster than existing IADS approaches. Also, it exhibits state-of-the-art performance on the MVTec AD dataset. Lastly, our approach also outperforms competitors in segmentation performance when considering a more challenging few-shot scenario built upon the VisA dataset.

A limitation of our method resides in the small spatial size of the output anomaly map, which is constrained by the leveraged backbone. An interesting future direction would be to employ strategies that can yield high-resolution feature maps such as FeatUp Fu et al. (2024).

## REPRODUCIBILITY STATEMENT

The main paper and Appendix contain all the details required to reproduce our work. Moreover, we will provide an anonymous GitHub link to our code in a private comment to reviewers in the discussion forum on OpenReview to ensure the reproducibility of our results. The code will be released publicly upon acceptance.

## ETHIC STATEMENT

We have not identified any ethical concerns related to our work.

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

# A    SUPPLEMENTAL MATERIAL

In this supplemental material, we provide additional quantitative and qualitative results to validate the performance of the proposed approach.

## A.1    ABLATION ON THE LAYERS CONSIDERED FOR THE FORWARD AND BACKWARD FEATURE TRANSFER NETWORKS.

We investigate the impact of transferring features from different levels of the transformer architecture, i.e., layers $j$ and $k$ described in Section 3, either by aggregating them or by considering the individual maps. In  Table 6, we report results for various combinations of layers. The notation $[j, k]$ means fusing both forward and backward transfer from layer $j$ to layer $k$ and vice-versa as seen in the main paper. With $[j \rightarrow k]$ or $[k \leftarrow j]$, we intend the performance of the individual anomaly map in a single direction. We note that transferring features between layers with high contextualization, i.e., the last four layers, begets better detection and segmentation results, with the transfer between $j = 8$ and $k = 12$ providing the best performance. We also observe that transferring features from closer layers, such as $j = 11$ and $k = 12$, can harm the performance. We believe that being the function learned by a single transformation layer smooth Jelenić et al. (2024), the task of transferring between two close layers is simpler. Thus, it might overgeneralize to anomalous samples, leading to worse performance. Conversely, between two farther layers, the function is highly non-linear. Nevertheless, the performance is relatively stable after layer 8, independent of the employed layers. Notably, fusing the maps obtained from the forward and backward transfer always yields the best results except for layers $[1, 4]$. We suggest that this occurs because the features in the earlier layers lack sufficient contextualization.

Table 6: **Layers Ablation.** Best results in **bold**, runner-ups underlined.

| LAYERS | I-AUROC | AUPRO@30% | AUPRO@5% |
|---|---|---|---|
| $[1, 4]$ | 0.906 | 0.828 | 0.570 |
| $[1 \rightarrow 4]$ | 0.913 | 0.906 | 0.682 |
| $[1 \leftarrow 4]$ | 0.773 | 0.663 | 0.378 |
| $[4, 8]$ | 0.940 | 0.941 | 0.773 |
| $[4 \rightarrow 8]$ | 0.924 | 0.942 | 0.764 |
| $[4 \leftarrow 8]$ | 0.931 | 0.903 | 0.702 |
| $[8, 12]$ | **0.964** | **0.952** | **0.787** |
| $[8 \rightarrow 12]$ | 0.953 | 0.943 | 0.773 |
| $[8 \leftarrow 12]$ | 0.949 | 0.925 | 0.745 |
| $[10, 12]$ | 0.960 | 0.950 | 0.784 |
| $[10 \rightarrow 12]$ | 0.957 | 0.926 | 0.742 |
| $[10 \leftarrow 12]$ | 0.960 | 0.947 | 0.782 |
| $[11, 12]$ | 0.956 | 0.946 | 0.774 |
| $[11 \rightarrow 12]$ | 0.868 | 0.853 | 0.710 |
| $[12 \rightarrow 11]$ | 0.888 | 0.876 | 0.730 |

## A.2    ABLATION ON THE LOSS EMPLOYED TO OPTIMIZE THE FORWARD AND BACKWARD FEATURE TRANSFER NETWORKS.

Table 7 reports the results obtained by the proposed framework considering different distances (i.e., cosine distance and $\ell_2$ distance) for the optimization and the inference of the forward and backward feature transfer networks, i.e., the Students MLPs. We report the results of the four possible combinations of these distances at training and inference time. Our chosen combination shows slightly better performance than the alternatives, though differences are minimal.

Table 7: **Loss ablation.** Best results in **bold**, runner-ups underlined.

| TRAINING | INFERENCE | I-AUROC | AUPRO@30% | AUPRO@5% |
|---|---|---|---|---|
| Cosine distance | $\ell_2$ distance | **0.964** | **0.952** | 0.787 |
| $\ell_2$ distance | $\ell_2$ distance | 0.954 | 0.950 | 0.786 |
| Cosine distance | Cosine distance | 0.957 | **0.952** | **0.790** |
| $\ell_2$ distance | Cosine distance | 0.954 | 0.938 | 0.741 |

### A.3 ABLATION ON THE FUNCTION EMPLOYED TO FUSE THE ANOMALY MAPS.

Given the best combination of transferring features between layers being between $j = 8$ and $k = 12$, as shown in Table 6, we also investigate the fusion strategy. In particular, we chose multiplication to minimize potential false positives from the maps produced by each student network. This operation can be viewed as a logical AND between the two maps, meaning that a pixel is categorized as a defect only if both student networks agree to predict it. As shown in Table 8, choosing multiplication as an aggregation function enhances the performance of the individual maps, while addition slightly degrades their performance.

Table 8: **Aggregation ablation.** $j = 8, k = 12$. Best results in **bold**, runner-ups underlined.

| ANOMALY MAP | I-AUROC | AUPRO@30% | AUPRO@5% |
|---|---|---|---|
| $\Delta_k \cdot \Delta_j$ | **0.964** | **0.952** | **0.787** |
| $\Delta_k + \Delta_j$ | 0.944 | 0.931 | 0.732 |
| $\Delta_j$ | 0.953 | 0.943 | 0.773 |
| $\Delta_k$ | 0.949 | 0.925 | 0.745 |

### A.4 IS DINO-V2 A GENERALLY BETTER IADS BACKBONE?

In Section 5, we demonstrated that DINO-v2 performs better with our approach, as it allows effective processing of high-resolution images. However, we also analyze whether the performance of other methods improves when using DINO-v2 with high-resolution inputs. Specifically, we evaluate two additional IADS methods—PatchCore Roth et al. (2022) and SPADE Cohen & Hoshen (2020)—which can easily accommodate changes to their backbone without requiring ad-hoc modifications. The results are reported in Table 9. As shown, neither PatchCore nor SPADE fully benefit from high-resolution processing, as their memory bank mechanisms do not scale well with increased resolution. Therefore, we conclude that DINO-v2 may not always be the best backbone for IADS.

Table 9: **Segmentation metrics on VisA.** Best results in **bold**, runner-ups underlined.

| ALGORITHM | BACKBONE | INPUT RESOLUTION | AUPRO@30% | AUPRO@5% |
|---|---|---|---|---|
| PatchCore | WideResNet101 | $224 \times 224$ | 0.779 | 0.508 |
| PatchCore | WideResNet101 | $512 \times 512$ | 0.752 | 0.542 |
| PatchCore | DINO-v2 | $1036 \times 1036$ | 0.705 | 0.445 |
| SPADE | WideResNet101 | $224 \times 224$ | 0.780 | 0.480 |
| SPADE | DINO-v2 | $1036 \times 1036$ | 0.779 | 0.462 |
| FBFT | ViT/B-16 | $224 \times 224$ | 0.868 | 0.610 |
| FBFT | DINO-v2 | $1036 \times 1036$ | **0.952** | **0.787** |

### A.5 ABLATION ON DIFFERENT INPUT RESOLUTION SIZES DURING TRAINING.

We report in Table 10 the anomaly detection and segmentation performance achieved by the proposed methods when different input resolution sizes are considered for the feature extractor. The same resolution is used during training and inference, while the evaluation is performed by upsampling the output to the original full resolution of the ground-truth. From these results, it is possible to appreciate that the proposed solution is able to exploit the higher resolution and correctly detect and segment the majority of samples, with an I-AUROC of 0.964 at full resolution, compared to 0.899

when providing low-resolution images. The same trend can be observed for the localization metrics, i.e., AUPRO@30% and AUPRO@5%.

Table 10: **Ablation on the input resolution employed at training time.** Best results in **bold**, runner-ups underlined.

| TRAINING RESOLUTION | I-AUROC | AUPRO@30% | AUPRO@5% |
|---|---|---|---|
| $224 \times 224$ | 0.899 | 0.830 | 0.562 |
| $518 \times 518$ | 0.952 | 0.944 | 0.779 |
| $1036 \times 1036$ | **0.964** | **0.952** | **0.787** |

### A.6 P-AUROC SEGMENTATION RESULTS ON VISA

We report in Table 11 the segmentation performance based on the P-AUROC metric on the VisA dataset alongside the other segmentation metrics. We note that our method also achieves state-of-the-art performance on this metric. We wish to highlight that even though is a common practice to evaluate the segmentation performance metric with such a metric, we believe that AUPRO@5% is the best metric to describe segmentation performance as it is less saturated (0.787 AUPRO@5% vs. 0.991P-AUROC for our method), and it considers each anomaly independently during calculation, making it suitable for our quartile-based evaluation.

Table 11: **Segmentation metrics on VisA.** Best results in **bold**, runner-ups underlined.

| ALGORITHM | P-AUROC | AUPRO@30% | AUPRO@5% |
|---|---|---|---|
| PatchCore | 0.902 | 0.752 | 0.542 |
| SimpleNet | 0.956 | 0.718 | 0.469 |
| EfficientAD | 0.977 | 0.937 | 0.777 |
| RD++ | 0.938 | 0.907 | 0.758 |
| FBFT (Ours) | **0.991** | **0.952** | **0.787** |

### A.7 MORE INSIGHT ON THE CONTEXTUALIZATION AND DECONTEXTUALIZATION TASKS

To better understand how these contextualization and decontextualization tasks are useful to detect anomalies, let us imagine that we are modelling images of nominal cats at training time. The Students (MLPs) have learned from the Teacher (Transformer) how a typical cat looks by understanding the relationships between the local features, like fur texture, whiskers, eyes, and the global features, like the overall shape and arrangement of body parts.

Then, at inference time we can distinguish four different scenarios:

**Nominal test sample.** Given a nominal test sample of a cat, during contextualization the Forward Network process the features of smaller details, such as fur texture, eye shapes, and ears and then correctly predicts the features of these details integrated into a broader context, realizing these features together form a coherent cat with proper body part arrangements. The Backward Network's global understanding of the cat is that it knows where the eyes, ears, and fur should be placed. When the Backward Network tries to map this global understanding back to local features, it succeeds because the local features match the global cat shape. Both contextualization and decontextualization succeed, confirming this is a nominal sample of a cat.

**Anomalous test sample that breaks both contextualization and decontextualization.** Given an anomalous sample, like a cat with bird wings, during contextualization, the Forward Network detect typical local cat features but also sees something odd, such as bird wings instead of the expected legs. Hence, when the Forward Network tries to build a global context, it struggles because bird wings do not fit into the overall cat structure. When trying to decontextualize, the Backward Network fails to map back correctly since the wings create confusion in its global representation and do not align with

the typical local features of a cat. Both Forward and Backward Networks detect this misalignment as an anomaly.

**Anomalous test sample that breaks only decontextualization.** Given an anomalous sample, like a cat with the fur texture subtly changed in some areas to resemble scales, the Forward Network processes local features, and since is still detecting the overall shape of the cat and other features, it forms a correct global understanding of the image as a whole, successfully building a global context. The overall structure of the cat is intact, so contextualization does not fail, since the cat still looks like a cat, even though some textures are unusual. However, during decontextualization, when the Backward Network tries to map the global context back to local features, the scale-like textures do not fit what the model expects from a cat's fur, breaking the consistency between the global understanding features and local textures features. The subtle anomaly did not disrupt the overall structure of the image, but when trying to map back to local details, the inconsistency in texture caused the model to fail. Only the Backward Network detect this misalignment as an anomaly.

**Anomalous test sample that breaks only contextualization.** Given an anomalous sample, like a cat where the head is slightly displaced, during contextualization, the Forward Network detects normal local features, however, when trying to form a global context, the misaligned cat's head leads to an incoherent global structure. Essentially, the parts of the cat are shifted slightly out of position, hence, the global context is broken but the local features are intact. Nevertheless, during decontextualization, although the global context is broken, the individual parts of the cat still seem coherent on their own. As a result, the decontextualization succeeds because the model can map back to the local features successfully, even though the global context was incorrect. Only the Forward Network detect this misalignment as an anomaly.

## A.8 Full results on VisA

For the sake of completeness, in Table 12 we report the per-class detection and segmentation performance, previously summarized in Table 1 of the main paper. Results of our solution and state-of-the-art methods on the VisA dataset are reported.

Table 12: **I-AUROC and AUPRO30@% on the VisA dataset for several IADS methods.** Best results in **bold**, runner-ups underlined. All methods are trained and tested at high-resolution.

| | ALGORITHM | candle | capsules | cashew | chewinggum | fryum | macaroni1 | macaroni2 | pcb1 | pcb2 | pcb3 | pcb4 | pipe_fryum | MEAN |
|---|---|---|---|---|---|---|---|---|---|---|---|---|---|---|
| **I-AUROC** | PatchCore Roth et al. (2022) | 0.986 | 0.937 | 0.990 | 0.991 | 0.993 | 0.997 | 0.934 | 0.980 | 0.988 | 0.996 | 0.998 | 0.998 | **0.982** |
| | SimpleNet Liu et al. (2023) | 0.964 | 0.769 | 0.972 | 0.984 | 0.922 | 0.809 | 0.618 | 0.984 | 0.956 | 0.949 | 0.937 | 0.986 | 0.904 |
| | EfficientAD Batzner et al. (2024) | 1.000 | 0.884 | 0.933 | 0.996 | 0.957 | 0.947 | 0.967 | 0.991 | 0.971 | 0.972 | 1.000 | 1.000 | 0.968 |
| | RD++ Tien et al. (2023) | 0.846 | 0.935 | 0.862 | 0.838 | 0.966 | 0.964 | 0.897 | 0.935 | 0.972 | 0.980 | 0.982 | 0.990 | 0.930 |
| | FBFT (Ours) | 0.958 | 0.992 | 0.972 | 0.996 | 0.988 | 0.931 | 0.885 | 0.980 | 0.938 | 0.956 | 0.976 | 0.999 | 0.964 |
| **AUPRO@30%** | PatchCore Roth et al. (2022) | 0.955 | 0.575 | 0.912 | 0.670 | 0.836 | 0.349 | 0.340 | 0.941 | 0.864 | 0.703 | 0.910 | 0.969 | 0.752 |
| | SimpleNet Liu et al. (2023) | 0.867 | 0.574 | 0.876 | 0.723 | 0.766 | 0.531 | 0.244 | 0.801 | 0.828 | 0.757 | 0.737 | 0.918 | 0.718 |
| | EfficientAD Batzner et al. (2024) | 0.982 | 0.897 | 0.888 | 0.822 | 0.895 | 0.968 | 0.982 | 0.945 | 0.948 | 0.950 | 0.982 | 0.982 | 0.937 |
| | RD++ Tien et al. (2023) | 0.964 | 0.959 | 0.699 | 0.642 | 0.919 | 0.977 | 0.979 | 0.932 | 0.938 | 0.957 | 0.949 | 0.967 | 0.907 |
| | FBFT (Ours) | 0.979 | 0.963 | 0.971 | 0.908 | 0.944 | 0.971 | 0.961 | 0.965 | 0.939 | 0.910 | 0.935 | 0.972 | **0.952** |
| **AUPRO@5%** | PatchCore Roth et al. (2022) | 0.823 | 0.402 | 0.783 | 0.491 | 0.497 | 0.140 | 0.133 | 0.799 | 0.609 | 0.399 | 0.592 | 0.831 | 0.542 |
| | SimpleNet Liu et al. (2023) | 0.660 | 0.396 | 0.650 | 0.444 | 0.397 | 0.221 | 0.121 | 0.611 | 0.579 | 0.385 | 0.715 | | 0.469 |
| | EfficientAD Batzner et al. (2024) | 0.897 | 0.675 | 0.715 | 0.582 | 0.585 | 0.839 | 0.897 | 0.779 | 0.775 | 0.782 | 0.897 | 0.897 | 0.777 |
| | RD++ Tien et al. (2023) | 0.859 | 0.826 | 0.505 | 0.384 | 0.749 | 0.872 | 0.879 | 0.789 | 0.811 | 0.827 | 0.739 | 0.851 | 0.758 |
| | FBFT (Ours) | 0.881 | 0.833 | 0.851 | 0.674 | 0.751 | 0.848 | 0.839 | 0.834 | 0.746 | 0.684 | 0.664 | 0.840 | **0.787** |

## A.9 Training time.

We provide in Table 13 the average time in hours needed per class to train every framework, given the number of epochs reported in their official implementations. These timings have been computed using the same hardware employed for all our experiments.

Table 13: **Training time required on the VisA dataset.** Average training time in hours per class. All methods are trained and tested at high resolution.

| ALGORITHM | PatchCore Roth et al. (2022) | SimpleNet Liu et al. (2023) | EfficientAD Batzner et al. (2024) | RD++ Tien et al. (2023) | FBFT (Ours) |
|---|---|---|---|---|---|
| Training time | 1.212 | 6.266 | 7.783 | 28.767 | 2.361 |

## A.10 IMPLEMENTATION EMPLOYED FOR THE COMPETITORS AND THEIR LICENSES

For all the competitors Roth et al. (2022); Liu et al. (2023); Tien et al. (2023), except Efficien-tAD Batzner et al. (2024), we employed their official implementations. As far as it concerts Efficien-tAD, which does not provide an official repository, we leverage an implementation that obtains the most similar results with respect to the values reported in their manuscript Batzner et al. (2024). In particular:

- PatchCore: `https://github.com/amazon-science/patchcore-inspection` released under Apache License 2.0;
- SimpleNet: `https://github.com/DonaldRR/SimpleNet` released under MIT License;
- RD++: `https://github.com/tientrandinh/Revisiting-Reverse-Distillation` released under MIT License;
- EfficientAD: `https://github.com/nelson1425/EfficientAD` released under Apache License 2.0.

## A.11 LICENSE FOR THE EMPLOYED DATASETS

The VisA dataset Zou et al. (2022) is released under the Creative Commons Attribution (CC BY 4.0) license. The MVTec AD dataset Bergmann et al. (2019) is released under the Creative Commons Attribution-NonCommercial-ShareAlike 4.0 International License (CC BY-NC-SA 4.0).

## A.12 ADDITIONAL QUALITATIVE RESULTS ON THE VISA AND MVTEC AD DATASETS.

As anticipated in the main paper, we show in Fig. 7 some additional qualitative results for the remaining classes of the VisA dataset which have not been reported in Fig. 5. As already highlighted in Section 5, the anomaly maps produced by our solution provide a more localized response for the anomalies, compared to EfficientAD Batzner et al. (2024).

Additionally, in Fig. 8 we show some qualitative examples of the anomaly map produced by our model on the MVTec AD dataset. Also in this scenario, our method provides more localized anomaly scores, motivating the segmentation performance gap in terms of both AUPRO@30% and AUPRO@5%.

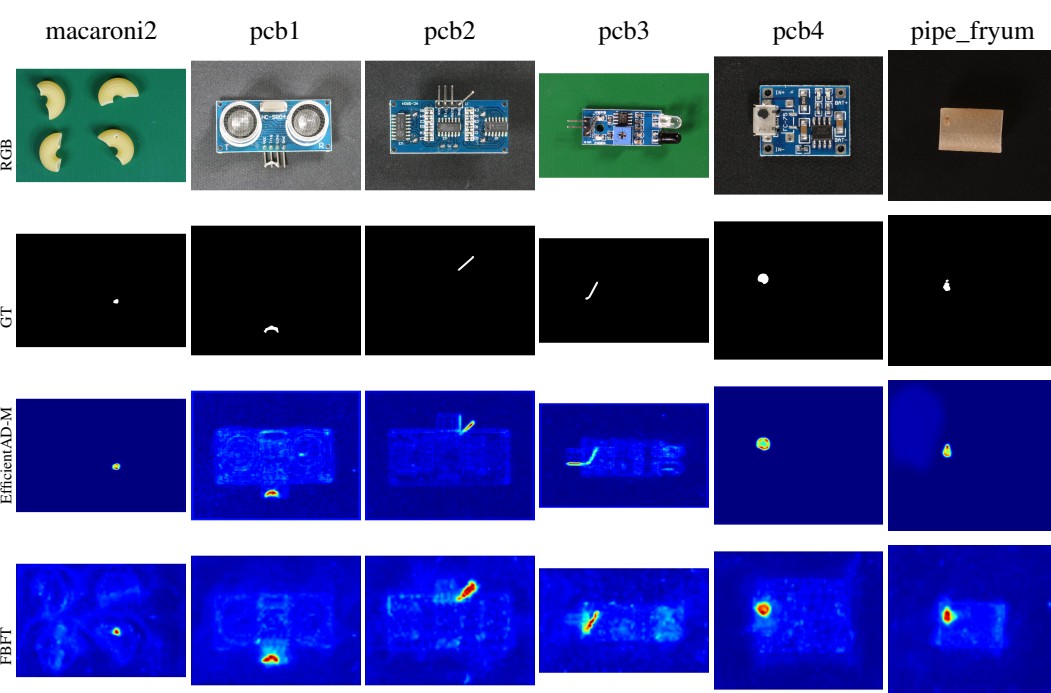

Figure 7: **VisA dataset qualitative results**. All methods are trained and tested at high resolution.

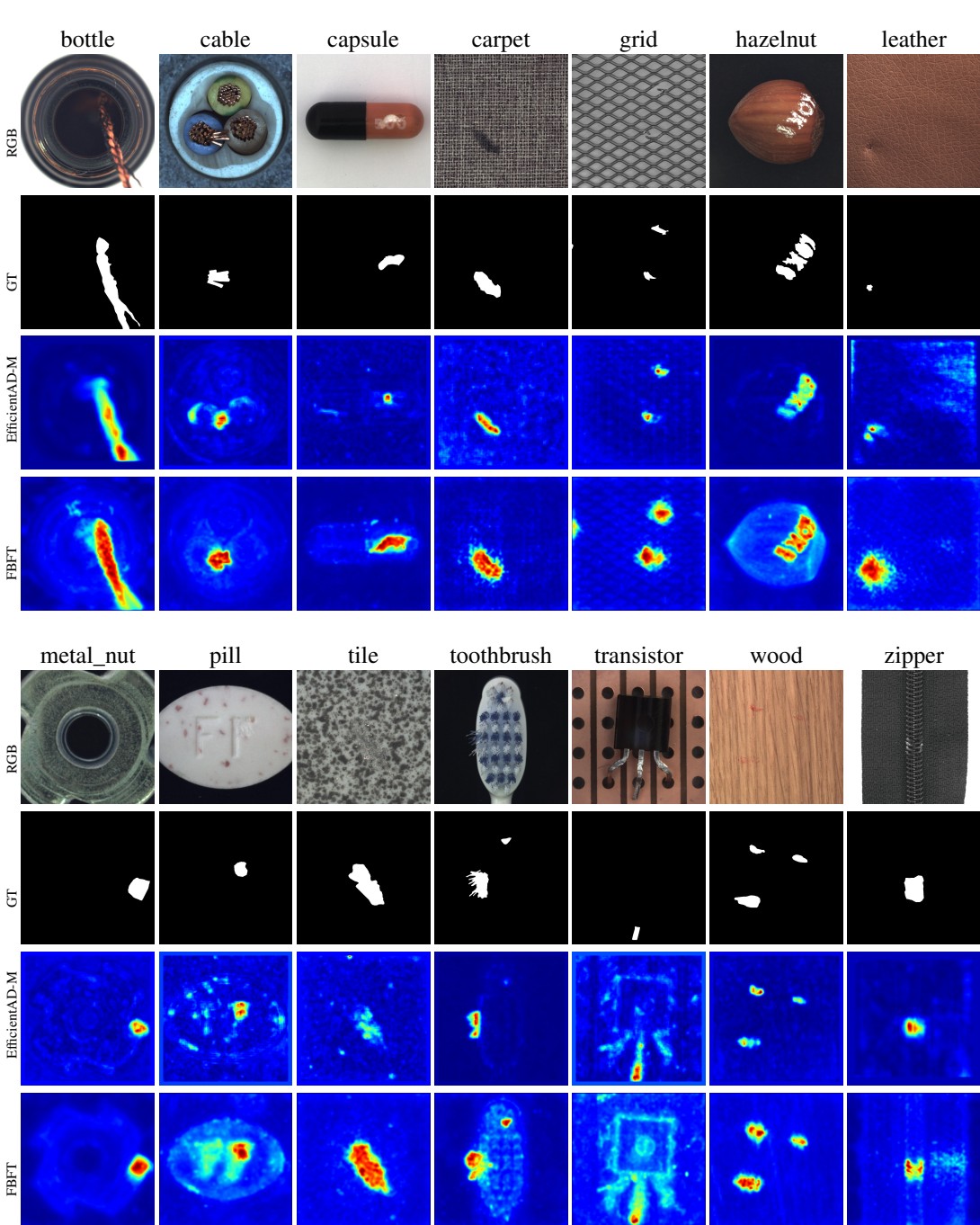

Figure 8: **MVTec AD dataset qualitative results**.

