# OpenReview forum: "Forward-Backward Feature Transfer for Industrial Anomaly Detection and Segmentation"
_ICLR.cc/2025/Conference — ICLR 2025 Conference Withdrawn Submission_

### Official Review · Reviewer_kYyx · 2024-10-23

**Soundness:** 2
**Presentation:** 2
**Contribution:** 2
**Rating:** 5
**Confidence:** 4

**Summary:**

This paper proposed an unsupervised method for anomaly detection. The method adopts a Teacher-Student architecture, where the Teacher is a frozen pre-trained transformer backbone network, and the Student consists of two networks, S_F and S_B. During training, the Student network is trained on nominal samples, where S_F is trained to predict features at deeper layer from features at shallow layer, and S_B vice versa. During inference, the difference between the Student predicted feature and Teacher extracted feature is used to generate anomaly map.

**Strengths:**

* The pretext task used to train the Student network, i.e., predict one layer's feature from another layer's feature is novel and interesting.
* The observation that previous methods downsample input image and ground truth for benchmarking is insightful, and the advocation that benchmarking should be done using ground truth at the original resolution makes sense.

**Weaknesses:**

* Apart from the pretext task to train the Student model, the overall framework, i.e., patch-level lightweight Teacher-Student model, is highly similar to EfficientAD. The advocation that benchmarking using ground truth at original resolution has also be done int EfficientAD. The other advocation, i.e., benchmarking using input image at original high resolution, may not always be necessary.
* The paper lacks benchmarking on the popular MVTec LOCO datasets.

**Questions:**

1. What is the structure of MLP used for the Student network and how does its design (e.g., number of layers, number of hidden units) affect the performance?
2. How does the method perform on logical anomaly detection, e.g., MVTec LOCO?
3. Please clarify why the proposed method is more efficient than EfficientAD [Batzner et al. 2024]. The design of the proposed method is highly similar to EfficientAD, and EfficientAD adopts more lightweight structure for the Teacher model. The major difference is that apart from the patch-level Teacher-Student model, EfficientAD additionally adopts an image-level autoencoder for logical anomaly detection. Is it because of the removal of the autoencoder that the proposed method is more efficient than EfficientAD? Would the removal of the autoencoder compromise the performance for logical anomaly detection?
4. There is disparity between  a method's performance reported in this paper and the performance reported in the method's original paper. For example, in Table 1, the paper report 0.965 I-AUROC for EfficientAD on MVTec AD, but in Table 2 of the EfficientAD paper  [Batzner et al. 2024], this number is 0.991. Please clarify.

---

### Official Review · Reviewer_7QJq · 2024-10-27

**Soundness:** 2
**Presentation:** 3
**Contribution:** 2
**Rating:** 5
**Confidence:** 4

**Summary:**

This paper introduces an efficient industrial anomaly detection and segmentation (IADS) approach using a novel Teacher-Student paradigm. The authors address the limitations of low-resolution image processing in detecting small defects. Their method uses two shallow Student MLPs to transfer features from a frozen Teacher Vision Transformer, allowing faster processing of high-resolution images with superior performance. They also propose new evaluation metrics that ensure stable defect detection across different sizes. The authors conducted extensive experiments to validate the effectiveness of this approach.

**Strengths:**

1.The authors address the limitations of low-resolution image processing in detecting small defects.
2.Their method uses two shallow Student MLPs to transfer features from a frozen Teacher Vision Transformer, allowing faster processing of high-resolution images with superior performance.
3.They also propose new evaluation metrics that ensure stable defect detection across different sizes. The authors conducted extensive experiments to validate the effectiveness of this approach.

**Weaknesses:**

The overall structural integrity of the paper and the completeness of the experiments are good, but I have a few minor questions:
1. The usage of the terms "nominal" and "hallucinate" in the paper is somewhat unclear. The author should clarify whether these terms are used appropriately.
2. From the perspective of method, the approach relies solely on a shallow MLP for cross-patch feature sharing, which while achieving faster speeds, does not provide substantial contributions in terms of network design. This approach is quite similar to previous methods based on distillation and reconstruction for one-class anomaly detection. The author cannot contextualize and decontextualize the abnormal region through the shared shallow MLP layer to identify abnormalities, which is itself a simplified version of feature reconstruction and teacher-student distillation.
3. In the evaluation protocol, the author mentions a checkpoint selection mechanism, noting that real-world scenarios may not have test samples available for model evaluation. However, some methods, such as RD and RD++, demonstrate a performance trend that first increases and then decreases as the number of training epochs grows. In contrast, the comparison methods used in the author's experiments select the final performance of the last epoch, which is unfair. Different models have varying numbers of parameters and performance trends, so comparing their performance with the author's model using, which optimizes only three linear layers, is unreasonable.
4. The experiments in the paper, as well as the supplementary materials, cover a broad and relatively complete range. However, the number of comparison methods is limited, and there is a lack of comparisons with similar methods, such as distillation and reconstruction approaches. Additionally, for a mainstream dataset like MVTec-AD, the paper does not compare more public metrics, including those proposed by the authors.
5. When resources allow, higher resolution generally improves anomaly detection performance. The authors have not provided sufficient evidence to demonstrate the advantages of their method beyond inference speed. They should include more comparisons of other classic methods with high-resolution inputs, including on MVTec-AD, VisA datasets, and metrics like P/I-AUROC, AP, and P-PRO. The faster speed of the method is only due to the use of three simple linear layers for distillation between Transformer layers.

**Questions:**

see weaknesses.

---

### Official Review · Reviewer_NG4N · 2024-10-29

**Soundness:** 2
**Presentation:** 3
**Contribution:** 1
**Rating:** 3
**Confidence:** 4

**Summary:**

The paper presents a novel method for Industrial Anomaly Detection and Segmentation (IADS) that addresses the limitations of current techniques which rely on low-resolution images and may miss smaller defects. This method introduces a forward-backward feature transfer technique utilizing a Teacher-Student paradigm where shallow MLPs (the Students) learn to mimic transformations across layers of a frozen Transformer (the Teacher). This approach allows for processing high-resolution images efficiently, improving the detection and localization of anomalies across various sizes.

**Strengths:**

High-Resolution Processing Capability: The novel unsupervised IADS approach enables the processing of high-resolution images, which allows for better detection of smaller anomalies, crucial in industrial contexts where anomalies vary greatly in size.

Efficiency: The method achieves faster anomaly detection compared to other methods, even when those methods are processing lower-resolution images. This is largely due to the use of shallow MLP student networks that process patch features independently, enabling fast batch processing.

**Weaknesses:**

Lack of innovation in methods.
The competitiveness of experimental results is limited.

**Questions:**

1. line53. It says "Student networks can hallucinate contextual information". Can a simple MLP learn context? Please provide experimental evidence or references to support this.

2. line150. "Forward and Backward Transfer Networks" Where does the method demonstrate Transfer? What is the difference between it and distillation?

3.line 311. when $AUPRO(Q_4)$ > $AUPRO(Q_1)$, $\rho = AUPRO_{averange} (AUPRO(Q_1)/(AUPRO(Q_4))$,
when $AUPRO(Q_1)$ > $AUPRO(Q_4)$, $\rho = AUPRO_{averange} (AUPRO(Q_4)/(AUPRO(Q_1))$, is this right?

4.Table1, Your I_AUROC gap is large,  I have doubts about the effectiveness of the method, please use more metrics for comparison, such as max-F1.

5. line449. The time for a single-image experiment is 1.786ms, does this time include backbone feature extraction?

6. Table5. The difference between ViT and Dino V2 is significant, indicating that the method is sensitive to the backbone. Please supplement other methods with the same backbone as Dino V2.

---

### Official Review · Reviewer_wFm5 · 2024-11-05

**Soundness:** 3
**Presentation:** 3
**Contribution:** 2
**Rating:** 5
**Confidence:** 5

**Summary:**

In order to solve the problem of large differences in defect sizes in IADS, this paper proposes a fast approach based on a novel Teacher-Student paradigm. A Transformer-based backbone is used as the Teacher, providing a set of features. Students are a pair of forward and backward transmission networks, each of which predicts the features of a layer based on the corresponding features extracted from the other layer and processes the position of each block independently. In addition, this paper proposes a novel evaluation metric to verify the robustness of the model to defects of different sizes.

**Strengths:**

The paper creatively proposes a forward and backward propagation mechanism to improve the recognition of small defects by fusing the results of anomaly detection at different layers. In addition, a new evaluation index is proposed, which provides a new perspective for evaluating the performance of anomaly detection models.

**Weaknesses:**

The paper lacks a detailed description of the internal structure of Students and the method of dividing defect sets of different sizes. In the comparative experiment of different algorithms,  the input images use different resolutions. Whether this will affect the fairness of the
 experimental results still needs to be explored.

**Questions:**

1.There is a lack of detailed explanation of the Student structure.
 2.In the Performance across defects sizes section of 4.2 metrics, what is the basis for the division
of Q1, Q2, Q3, and Q4?
 3.In the Anomaly detection and segmentation experiment, different methods use different
resolutions. Are their results comparable? Will it affect the fairness of the experimental results?
 4.It is recommended to add ablation experiments in Table5 that use the same backbone as FBFT
and other algorithms.
 5.There are several spelling errors throughout the paper and the author is advised to perform a
comprehensive spell check

---

### Note · Authors · 2024-11-12

I have read and agree with the venue's withdrawal policy on behalf of myself and my co-authors.